# Beta Random Restart Strategy-based Remora Optimization Algorithm for Global Optimization

1st Zekai Ai
*College of Design and Engineering*
*National University of Singapore*
Singapore
aizekai@u.nus.edu

2nd Xiaoming Shi *
*School of Information Engineering*
*Sanming University*
Sanming, China
20210868130@fjsmu.edu.cn

3rd Heming Jia *
*School of Information Engineering*
*Sanming University*
Sanming, China
jiaheming@fjsmu.edu.cn

4th Jie Yang
*School of Information Engineering*
*Sanming University*
Sanming, China
20210852147@fjsmu.edu.cn

5th Bowen Xue
*School of Electrical and Information Engineering*
*Northeast Petroleum University*
Daqing, China
xuebowen@stu.nepu.edu.cn

6th Yilong Du
*College of Information and Electrical Engineering*
*Heilongjiang Bayi Agricultural University*
Daqing, China
bayi15093488812@byau.edu.cn

*Abstract* *Abstract*—The Remora Optimization Algorithm (ROA) is a meta-heuristic algorithm that imitates the foraging behaviors of Remora. Its main idea lies in simulating the mechanism of switching hosts during the foraging process of remora. Due to the randomness of remora host selection, ROA frequently gets trapped in local optima, which slows down its convergence speed. To develop a more robust algorithm, this paper simulates the exploration and elimination mechanism in the biological evolution process and improves the random restart strategy with "prior" properties. Beta Random Restart Strategy-based Remora Optimization Algorithm (BROA) is proposed to realize global optimization. The proposed Beta-random restart strategy ranks individuals by fitness and restarts the less fit half. The Cauchy strategy is used to expand the search area and enhance the ability to escape from local optima. This paper meticulously assesses the performance of BROA using five comparison algorithms. Firstly, the optimization capabilities of BROA are assessed through CEC2020 tests. The Wilcoxon test assesses the difference between BROA and five different algorithms. Finally, Cantilever Beam Design problem is used for testing the practicability of BROA. Comprehensive results show that BROA performs best in CEC2020 and the cantilever beam design problem compared with five different optimization algorithms.

*Keywords-remora optimization algorithm, beta random restart strategy, Cauchy strategy, metaheuristics, swarm intelligence*

## I. INTRODUCTION

Meta-heuristic algorithms (MAs) are high-level strategies designed to tackle complex optimization problems, particularly in scenarios where traditional optimization algorithms may fail or perform inadequately [1]. Meta-heuristic algorithms are classified into three types: evolution-based algorithms, such as Genetic Algorithms (GA) [2], which simulate the natural genetic and selection processes, and Differential Evolution (DE) [3]; swarm-based algorithms, such as Particle Swarm Optimization Algorithm [4] which represents origin of the swarm intelligent algorithm, Reptile Search Algorithm (RSA) [5] which simulates the reptiles moving mode, Marine Predators Algorithm (MPA) [6] and Crayfish Optimization Algorithm (COA) [7]; and physics-based algorithms, such as the Gravitational Search Algorithm (GSA) [8]. Remora Optimization Algorithm (ROA) [9] is derived from the foraging behavior of Remora. The inspiration of ROA is to perform behaviors such as host switching according to the phenomenon that Remora will choose different strategies at different times when foraging. ROA is the same as most algorithms. In the early search, ROA often misses the global optimum, falling into local optima, and its convergence speed requires improvement. Many studies have been conducted to enhance these characteristics. Wang et al. [10] applied the roulette wheel selection and chaotic tent mapping to improve ROA. Wen et al. [11] used many strategies to improve ROA. The methods employed encompassed both the reverse joint opposite selection technique and the restart method. Wang et al. [12] improved ROA through the restart strategy and adaptive dynamic probability and developed an enhanced version with superior optimization capabilities. However, the exploration capacity of ROA has not received sufficient attention, which leads to the slow convergence speed of ROA. Moreover, the inherent randomness of the roulette wheel selection process may cause the algorithm to overlook the optimal solution while searching. Therefore, this paper first uses the beta distribution random number with a "prior" nature to propose an improved Beta random restart strategy, which simulates the exploration and elimination of the biological evolution process and improves the algorithm by avoiding the search process being trapped in the local best solution. In this paper, the Beta Random Restart Strategy-based Remora Optimization Algorithm (BROA) is presented to solve the problem that the best solution often falls into local optimality. This work's primary contributions can be outlined as follows:

1) The proposed solution improves the conventional random restart strategy which significantly increases overall computational cost, simulates the exploration of biology, and proposes a Beta random restart strategy with a 'prior' nature, effectively improving the algorithm's convergence speed.

2) To enhance the effectiveness of the Beta random restart strategy, the Cauchy algorithm is incorporated to increase disturbance and facilitate escaping local optima.

The following parts of this paper are presented below: Section II covers the conventional ROA. Section III illustrates the Beta random restart and Cauchy strategies. Section IV discusses the experimental results and compares BROA with various algorithms. Section V applies BROA to the cantilever beam design problem, comparing it with other algorithms to showcase its advantages.

## II. REMORA OPTIMIZATION ALGORITHM

The ROA's principle is similar to the foraging behavior of remoras. First, the remora attaches to a selected host to get the necessary food and switches between swordfish and whales using its experience. At the same time, to get food more effectively and quickly, the remora will also perform experience attacks and host feeding. The Sailfish Optimizer (SFO) algorithm will be used in the exploration stage, and Whale Optimization Algorithm (WOA) will be used in the exploitation stage. Figure 1 shows the foraging process.

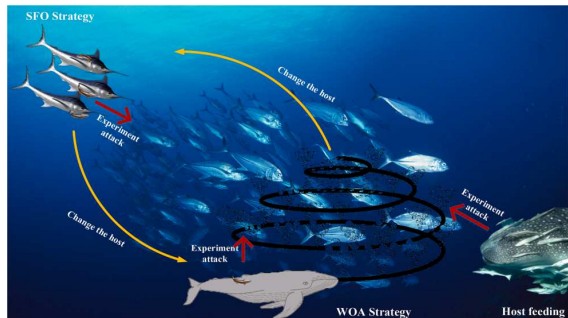

Figure 1.   A schematic diagram of the foraging process of remoras. [9]

### A. Initialization

At the initialization phase, random solutions are created within the search space, with each serving as the starting point for the ROA solution, as detailed in Formula (1):

$$X_i = lb + \text{rand} \times (ub - lb) \qquad (1)$$

Where $X_i$ is the remora's position, $lb$ and $ub$ define the lower and upper bound.

### B. Exploration

#### 1) SFO Strategy

The SFO algorithm is used to update the position of the sailfish. The parade speed is faster in the ROA algorithm's exploration stage, and the remora will attach itself to the sailfish and follow it as it swims. This process is shown in Formula (2):

$$X_i^{t+1} = X_{Best}^{t} - (\text{rand} \times (\frac{X_{Best}^{t} + X_{rand}^{t}}{2}) - X_{rand}^{t}) \qquad (2)$$

#### 2) Experience attack strategy

The remora performs localized, experience-based movements by taking the host's position and the previous position into consideration. The revised formula is given in Equation (3):

$$X_{att} = X_i^{t} + (X_i^{t} - X_{pre}) \times \text{randn} \qquad (3)$$

Where $X_{att}$ is the remora's exploratory motion attack, $X_{pre}$ indicates the remora's position in the previous generation.

After multiple attempts, Equation (4) decides if the hosts of remoras need to be changed. The equation for the remoras to switch the hosts is given in Formula (5).

$$f(X_i^{t}) < f(X_{att}) \qquad (4)$$

$$G(i) = \text{round}(\text{rand}) \qquad (5)$$

### C. Exploitation

#### 1) WOA Strategy

During the exploration stage of the ROA algorithm, the remora attaches itself to the whale and moves along with it. The updating position formula for the remora is derived from the Whale Optimization Algorithm (WOA). Revised formulas can be found in Equations (6)-(9):

$$X_i^{t+1} = D \times e^k \times \cos(2\pi a) + X_i^{t} \qquad (6)$$

$$D = \left| X_{Best}^{t} - X_i^{t} \right| \qquad (7)$$

$$k = \text{rand} \times (a - 1) + 1 \qquad (8)$$

$$a = -(1 + \frac{t}{T}) \qquad (9)$$

Where D measures the separation distance from the best individual's position to that of the current individual. The parameter $\alpha$ decreases within the range [-2, -1], and $T$ denotes the maximum times of iterations.

#### 2) Host Feeding Process

During the exploration stage of the ROA, remoras concentrate on a smaller area around the host. Equations (10)-(13) outline the location update formulas:

$$X_i^{t+1} = X_i^{t} + A \qquad (10)$$

$$A = B * (X_i^{t} - C * X_{Best}) \qquad (11)$$

$$B = 2 * V * \text{rand} - V \qquad (12)$$

Identify applicable sponsor/s here. *(sponsors)*

$$V = 2*(1-\frac{t}{T}) \qquad (13)$$

A indicates the distance from the remora's previous and current position. The constant C is set at 0.1 to constrain the remora's position. B represents random host volume space.

## III. PROPOSED METHOD

### A. Beta random restart strategy

Conventional restart strategies restart only after a set number of iterations, making them ineffective at promptly addressing "bad" individuals that hinder the progress of "good" ones. This can result in too many "bad" individuals, causing that the algorithm will focus on the local best solution but overlook global best solution. In order to effectively deal with the shortcomings of the traditional restart strategy, the Beta random restart strategy is proposed with a "prior" property. If the iterations surpass $T_{beta}$, the algorithm ranks individuals by fitness and restarts the less fit half. This ensures that the "bad" individuals can be restarted and refreshed, ensuring the BROA's fast and accurate convergence. Additionally, the random iteration count increases population diversity, boosts the exploratory capabilities of the conventional ROA, and helps avoid being stuck in local optima. The equations of the Beta restart strategy are as follows:

$$T_{beta} = Beta(\alpha, \beta)*T \qquad (14)$$

$$\alpha = 5*rand \qquad (15)$$

$$\beta = 5*rand \qquad (16)$$

$$X_i^{inferior} = lb + (ub - lb)*rand \qquad (17)$$

Where $T_{beta}$ denotes the times of the beta random restart, $X_i^{inferior}$ expresses the new position of the individual $i$. Term $Beta(\alpha,\beta)$ presents a random number that follows Beta distribution.

### B. Cauchy strategy

To widen the searching area and enhance the ROA's escape ability from local optima, it's important to address the relatively low peak value of the Cauchy distribution function to mitigate its impact on the search process. Cauchy mutation is used to increase variety in the population, thereby broadening the range of the ROA distribution. The local optimum is jumped out and the solution will be much closer to the global best solution.

$$ewbest = X_G + 0.05*X_G*cauchy(0,1) \qquad (18)$$

The pseudocode of BROA is presented below:

## Algorithm 1. BROA pseudocode

Initializing parameters

Use Formula (1) to initialize the population

While ($t < T$)
  Restrict the search space within the boundary.

  Evaluate each individual's fitness and get the new $X_{best}$

  Beta random restart strategy by Formula (14) - (17)

  If $rand > 0.5$

    Cauchy strategy by Formula (18)

  End if

  For individual $i$ from 1 : $n$

    If $G(i)$ equals to 0

      Formula (2) is used to get the new position.

    Else if $G(i)$ equals to 1
      Using Formula (6) - (9) to get the individual's position.

    End if

    Experience attack, as Equation (3)

    If $f(X_{att})$ is smaller than $f(X_i^t)$
      Switching the hosts of Remoras to get the new $X_i^t$

    Else
      Remora's host feeding as the Equation (10) to Equation (13)
    End if
  End for
  $t = t + 1$
End
Return $X_{best}$

## IV. EXPERIMENT ANALYSIS

CEC 2020 has 10 test functions. The CEC1 function is unimodal. The CEC3-CEC4 functions are basic. The CEC5-CEC7 functions are hybrid, and the CEC8-CEC10 functions in CEC 2020 are composition functions.

### A. Environment and parameters setting

All tests were conducted on MATLAB R2022a on a PC with AMD Ryzen™ 7 5800H CPU @3.20 GHz on OS Windows 11. In this paper, the proposed BROA is conducted a comparison with the conventional ROA and 4 different algorithms, including Whale Optimization Algorithm (WOA) [13], Arithmetic Optimization Algorithm (AOA) [14], Sine Cosine Algorithm (SCA) [15], and Spotted Hyena Optimizer (SHO) [16]. All algorithms follow the input of 30 population size and 500 maximum iterations and run 100 times independently. Moreover, the parameter settings in the experiment for these 6 algorithms are shown below:

TABLE I. PARAMETER SETTING

| Algorithm | Parameters |
|---|---|
| BROA | $C=0.1, C1=0.2$ |
| ROA | $C=0.1$ |
| WOA | $B = 1, a1 = [2,0], a2 = [-2,-1]$ |
| AOA | $a = 5, \mu = 0.5$ |
| SCA | $a = 2$ |
| SHO | $u=0.05, v=0.05, l=0.05$ |

## B. Comparison results

For CEC2020, we can see from Table II that BROA performs better optimization in 10 test functions. In CEC1 and CEC4-CEC10, BROA performs better than all comparison algorithms. In CEC2 and CEC3, the simplicity of the test functions allows most algorithms to locate the optimal value. From TABLE II, we can see that BROA has excellent stability under different conditions. In Figure 2, BROA performs well. For CEC1 - CEC10, BROA shows strong convergence ability. BROA can always perform stably, whether it is a simple or complex function.

TABLE II. EXPERIMENT RESULTS OF CEC2020(DIM=10)

| Function | index | BROA | ROA | WOA | AOA | SCA | SHO |
|---|---|---|---|---|---|---|---|
| CEC2020-1 | Best | **8.25E+05** | 1.36E+09 | 9.16E+06 | 4.65E+09 | 6.19E+08 | 9.29E+09 |
| | Mean | **1.79E+07** | 5.70E+09 | 1.04E+08 | 1.08E+10 | 1.06E+09 | 1.69E+10 |
| | Std | **1.86E+07** | 3.98E+09 | 1.47E+08 | 4.25E+09 | 3.97E+08 | 4.95E+09 |
| CEC2020-2 | Best | **1.37E+03** | 2.11E+03 | 1.92E+03 | 1.98E+03 | 2.20E+03 | 3.24E+03 |
| | Mean | **2.05E+03** | 2.47E+03 | 2.30E+03 | 2.26E+03 | 2.58E+03 | 3.54E+03 |
| | Std | 2.63E+02 | 3.05E+02 | 3.58E+02 | 3.17E+02 | **2.18E+02** | 2.97E+02 |
| CEC2020-3 | Best | **7.24E+02** | 7.75E+02 | 7.57E+02 | 7.80E+02 | 7.68E+02 | 8.45E+02 |
| | Mean | **7.68E+02** | 8.15E+02 | 8.00E+02 | 8.03E+02 | 7.86E+02 | 8.63E+02 |
| | Std | 1.49E+01 | 3.01E+01 | 3.43E+01 | 1.28E+01 | **1.14E+01** | 2.61E+01 |
| CEC2020-4 | Best | **1.90E+03** | **1.90E+03** | **1.90E+03** | **1.90E+03** | **1.90E+03** | **1.90E+03** |
| | Mean | **1.90E+03** | **1.90E+03** | 1.90E+03 | **1.90E+03** | 1.90E+03 | **1.90E+03** |
| | Std | **0.00E+00** | **0.00E+00** | 4.40E-01 | **0.00E+00** | 1.81E+00 | **0.00E+00** |
| CEC2020-5 | Best | **2.43E+03** | 8.44E+03 | 7.54E+03 | 1.07E+05 | 1.51E+04 | 8.43E+05 |
| | Mean | **1.61E+04** | 4.73E+05 | 4.99E+05 | 4.22E+05 | 1.17E+05 | 6.19E+06 |
| | Std | **2.07E+04** | 3.50E+05 | 1.03E+06 | 2.24E+05 | 1.37E+05 | 6.73E+06 |
| CEC2020-6 | Best | **1.60E+03** | 1.76E+03 | 1.70E+03 | 1.88E+03 | 1.75E+03 | 2.22E+03 |
| | Mean | **1.81E+03** | 1.97E+03 | 1.87E+03 | 2.13E+03 | 1.86E+03 | 2.63E+03 |
| | Std | **8.93E+01** | 1.56E+02 | 1.36E+02 | 2.25E+02 | 9.33E+01 | 3.25E+02 |
| CEC2020-7 | Best | **2.41E+03** | 3.17E+03 | 1.38E+04 | 4.21E+03 | 5.52E+03 | 4.46E+05 |
| | Mean | **9.73E+03** | 3.61E+05 | 1.56E+06 | 1.56E+06 | 1.70E+04 | 5.78E+06 |
| | Std | **8.36E+03** | 1.13E+06 | 3.60E+06 | 2.96E+06 | 1.16E+04 | 7.27E+06 |
| CEC2020-8 | Best | **2.23E+03** | 2.38E+03 | 2.31E+03 | 2.69E+03 | 2.35E+03 | 3.20E+03 |
| | Mean | **2.31E+03** | 2.73E+03 | 2.62E+03 | 3.08E+03 | 2.49E+03 | 3.98E+03 |
| | Std | **1.81E+01** | 3.48E+02 | 5.77E+02 | 3.61E+02 | 3.27E+02 | 5.66E+02 |
| CEC2020-9 | Best | **2.50E+03** | 2.71E+03 | 2.75E+03 | 2.77E+03 | 2.78E+03 | 2.87E+03 |
| | Mean | **2.76E+03** | 2.82E+03 | 2.79E+03 | 2.85E+03 | 2.78E+03 | 2.96E+03 |
| | Std | **4.98E+01** | 6.18E+01 | 5.37E+01 | 8.95E+01 | 5.28E+01 | 8.43E+01 |
| CEC2020-10 | Best | **2.63E+03** | 2.99E+03 | 2.93E+03 | 3.10E+03 | 2.95E+03 | 3.43E+03 |
| | Mean | **2.94E+03** | 3.23E+03 | 2.99E+03 | 3.45E+03 | 2.99E+03 | 3.85E+03 |
| | Std | **2.27E+01** | 2.23E+02 | 1.19E+02 | 2.80E+02 | 2.78E+01 | 3.60E+02 |

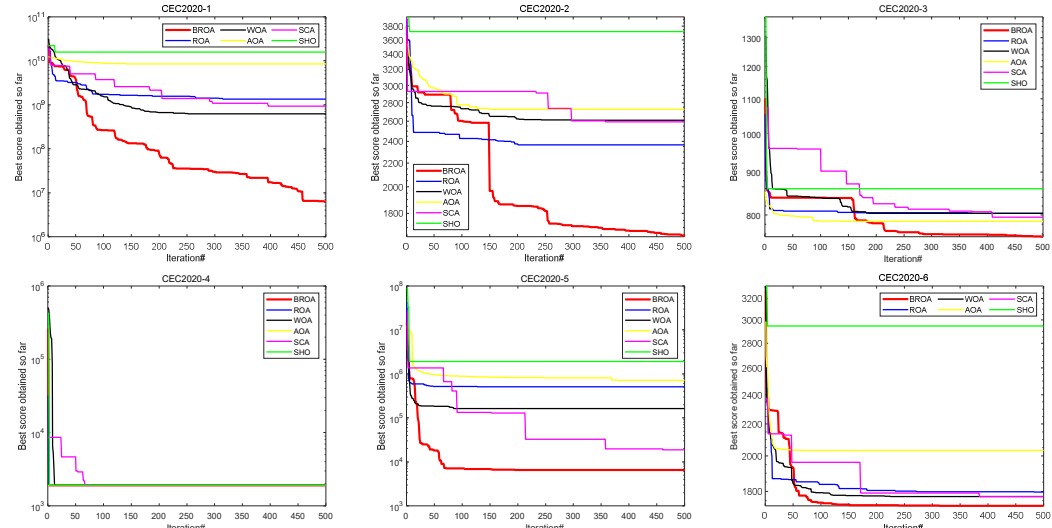

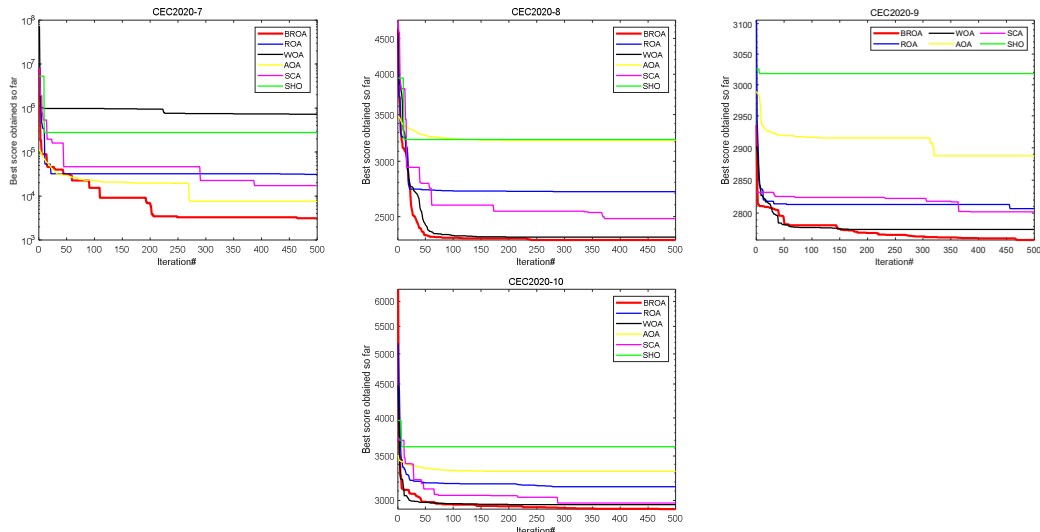

Figure 2.   Convergence diagram of CEC2020 test function.

## C.   Wilcoxon experiment

To analyze the effectiveness of the test experimental results of CEC2020, this subsection carries out the Wilcoxon rank-sum experiment. TABLE III represents the inspection results of CEC2020. Among them, data less than 0.05 are shown in bold. As demonstrated in TABLE III, most of the test results are less than 5%, indicating that BROA has significant differences from other algorithms and that BROA has enhanced optimization performance concerning the other 5 optimization algorithms.

TABLE III.       THE WILCOXON TEST DATA OF CEC 2020 TEST FUNCTION.

| CEC2020 | BROA VS ROA | BROA VS AOA | BROA VS SCA | BROA VS SHO | BROA VS WOA |
|---------|-------------|-------------|-------------|-------------|-------------|
| CEC1 | **1.73E-06** | **1.73E-06** | **2.83E-04** | **1.73E-06** | **7.73E-03** |
| CEC2 | **2.22E-04** | **4.73E-06** | **1.96E-02** | **2.60E-06** | **2.35E-06** |
| CEC3 | **1.92E-06** | **1.73E-06** | **3.32E-04** | **3.88E-04** | **1.73E-06** |
| CEC4 | 1.00E+00 | **1.73E-06** | **1.56E-02** | **8.86E-05** | **2.56E-06** |
| CEC5 | **1.73E-06** | **1.73E-06** | **1.97E-05** | **1.15E-04** | 6.87E-02 |
| CEC6 | **1.06E-04** | **3.72E-05** | **4.53E-04** | **1.83E-03** | **1.60E-04** |
| CEC7 | **8.31E-04** | **1.73E-06** | **1.73E-06** | **3.06E-04** | **1.11E-02** |
| CEC8 | **1.73E-06** | **1.73E-06** | **1.20E-03** | **1.73E-06** | **5.29E-04** |
| CEC9 | **1.49E-05** | **5.22E-06** | **1.71E-03** | **2.60E-05** | **1.15E-04** |
| CEC10 | **1.73E-06** | **1.15E-04** | **6.32E-05** | **3.18E-06** | **1.48E-02** |

## V.   CANTILEVER BEAM DESIGN PROBLEM

This section further validates the BROA algorithm's performance by applying it to a real-world engineering design problem, demonstrating its effectiveness in a practical engineering scenario.

The objective of this problem is to use the optimization algorithm to determine the relevant decision variables and ultimately reduce the total weight of the square section beam, thereby improving the engineering problem. Graph 3 presents an introduction of this engineering problem model.

The formulas for this problem are as follows:

Consider:

$$x = [x_1 \ x_2 \ x_3 \ x_4 \ x_5] \tag{19}$$

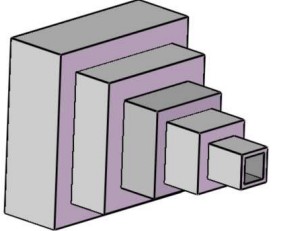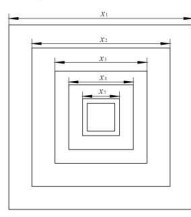

Figure 3.   Structural diagram of Cantilever beam design.

Objective function:

$$f(x) = 0.0624(x_1 + x_2 + x_3 + x_4 + x_5) \tag{20}$$

Subject to:

$$g(x) = \frac{61}{x_1^3} + \frac{37}{x_2^3} + \frac{19}{x_3^3} + \frac{7}{x_4^3} + \frac{1}{x_5^3} - 1 \le 0 \qquad (21)$$

Boundaries:

$$0.01 \le x_i \le 100 (i = 1, 2, \cdots 5) \qquad (22)$$

The statistical data table for the cantilever design experiment is presented below. The values of $x_i$ (where $i$=1,2,3,4,5) obtained by the BROA algorithm show a gradual decrease, which aligns with the cantilever beam's design principles. This results in an optimal weight of 1.34048732090604. Compared to the experimental data from other algorithms, the BROA algorithm delivers better performance in this problem.

TABLE IV. EXPERIMENT RESULTS OF CANTILEVER BEAM DESIGN.

| Algorithm | Optimal Values for Variables | | | | | Optimum Weight |
|---|---|---|---|---|---|---|
| | $x_1$ | $x_2$ | $x_3$ | $x_4$ | $x_5$ | |
| BROA | **6.14E+00** | **5.31E+00** | **4.46E+00** | **3.44E+00** | **2.13E+00** | **1.34E+00** |
| ROA | 6.86E+00 | 4.91E+00 | 5.86E+00 | 2.69E+00 | 2.86E+00 | 1.45E+00 |
| AOA | 6.36E+00 | 1.07E+01 | 5.35E+00 | 2.61E+00 | 2.41E+00 | 1.71E+00 |
| HHO | 6.14E+00 | 5.14E+00 | 4.59E+00 | 3.47E+00 | 2.15E+00 | 1.34E+00 |
| SOA | 6.00E+00 | 5.31E+00 | 4.47E+00 | 3.51E+00 | 2.21E+00 | 1.34E+00 |
| SCA | 5.82E+00 | 5.43E+00 | 4.80E+00 | 4.02E+00 | 1.87E+00 | 1.37E+00 |
| SHO | 6.23E+00 | 5.82E+00 | 4.34E+00 | 3.30E+00 | 2.59E+00 | 1.39E+00 |

## VI. CONCLUSION

This paper introduces a modified version of ROA. In the early search, ROA easily misses the global best solution, thus just focusing on the local best solution, and its convergence speed is insufficient. To solve above problems and strengthen the optimization ability of ROA, this paper proposes strategies to improve the conventional random restart strategy and applies the Beta random restart strategy and Cauchy strategy to strengthen this algorithm's developing capacity and help to leave the local optimal solution. In the experimental section, CEC2020 test functions are used for assessing the performance of BROA, compared with the conventional ROA and 4 other different algorithms. The results show that BROA performs excellently. Finally, an engineering problem is used to verify the engineering practicability of BROA. In future work, we will enhance BROA and apply it to more fields, such as UAV three-dimensional path planning problems, intrusion detection, medical resource scheduling problems, feature selection, etc.

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
