# OpenReview forum: "Beta Random Restart Strategy-based Remora Optimization Algorithm for Global Optimization"
_IEEE.org/ICIST/2024/Conference — IEEE ICIST 2024 Conference Submission_

### Official Review · Reviewer_WfJN · 2024-08-21
**Accept**

**Rating:** 7
**Confidence:** 3

**Review:**

This paper presents an intriguing enhancement to the Remora Optimization Algorithm (ROA) by addressing one of its key limitations—its tendency to get trapped in local optima due to the randomness in host selection during the foraging process. By introducing the Beta Random Restart Strategy (BROA), the authors have developed a more robust version of ROA that enhances global optimization capabilities. This improvement is rooted in simulating biological evolution processes, particularly focusing on exploration and elimination mechanisms, which is a creative approach to overcoming the original algorithm’s limitations. The reviewer has the following questions to discuss with the authors:

1. What are the "prior" properties in the Beta Random Restart Strategy, and how do they contribute to the improved global optimization capabilities of BROA?

2. How does the randomness in host selection specifically contribute to ROA getting trapped in local optima? Are there scenarios where this randomness could be beneficial?

3. What are the five comparison algorithms used in the study, and why were they chosen? How do they differ from BROA in terms of underlying mechanisms?

---

### Official Review · Reviewer_26ko · 2024-08-22
**Enhance work**

**Rating:** 6
**Confidence:** 3

**Review:**

The paper presents a promising enhancement to the Remora Optimization Algorithm through the introduction of a Beta random restart strategy and Cauchy strategy, aiming to improve global optimization capabilities. While the abstract effectively communicates the key contributions and methodological rigor of the study, it could benefit from clearer explanations of technical terms, more detailed comparative insights, and a discussion of the broader implications of the research. The following comments need to further consider: a. Provide a brief explanation of the Beta random restart strategy and the Cauchy strategy within the abstract. b. Mention specific results or findings from the comparison with other algorithms, highlighting how BROA stands out in terms of performance, efficiency, or other relevant metrics. c. Discuss the potential broader applications of BROA beyond the Cantilever Beam Design problem. This could include other engineering problems, data analysis tasks, or any field where global optimization is critical.

---

### Official Review · Reviewer_Xg8M · 2024-08-22
**Manuscript Accept**

**Rating:** 7
**Confidence:** 4

**Review:**

What is the specific advantage of the 'prior' property?
How do the authors ensure that the global optimization is achieved?
Fig. 2 shows the better convergence of BROA in the CEC2020 test function. Will it bring different convergence results when more or fewer iteration numbers are set?

---

### Decision · Program_Chairs · 2024-09-08

Accept (Oral)